# Flotillins: At the Intersection of Protein *S-*Palmitoylation and Lipid-Mediated Signaling

**DOI:** 10.3390/ijms21072283

**Published:** 2020-03-26

**Authors:** Katarzyna Kwiatkowska, Orest V. Matveichuk, Jan Fronk, Anna Ciesielska

**Affiliations:** 1Laboratory of Molecular Membrane Biology, Nencki Institute of Experimental Biology of the Polish Academy of Sciences, 3 Pasteur St., 02-093 Warsaw, Poland; o.matveichuk@nencki.edu.pl (O.V.M.); a.ciesielska@nencki.edu.pl (A.C.); 2Institute of Biochemistry, Faculty of Biology, University of Warsaw, 2 Miecznikowa St., 02-096 Warsaw, Poland; fronk@biol.uw.edu.pl

**Keywords:** flotillin, phosphatidylinositol, rafts, *S-*palmitoylation, sphingosine

## Abstract

Flotillin-1 and flotillin-2 are ubiquitously expressed, membrane-associated proteins involved in multifarious cellular events from cell signaling, endocytosis, and protein trafficking to gene expression. They also contribute to oncogenic signaling. Flotillins bind the cytosolic leaflet of the plasma membrane and endomembranes and, upon hetero-oligomerization, serve as scaffolds facilitating the assembly of multiprotein complexes at the membrane–cytosol interface. Additional functions unique to flotillin-1 have been discovered recently. The membrane-binding of flotillins is regulated by *S-*palmitoylation and *N-*myristoylation, hydrophobic interactions involving specific regions of the polypeptide chain and, to some extent, also by their oligomerization. All these factors endow flotillins with an ability to associate with the sphingolipid/cholesterol-rich plasma membrane domains called rafts. In this review, we focus on the critical input of lipids to the regulation of the flotillin association with rafts and thereby to their functioning. In particular, we discuss how the recent developments in the field of protein *S-*palmitoylation have contributed to the understanding of flotillin1/2-mediated processes, including endocytosis, and of those dependent exclusively on flotillin-1. We also emphasize that flotillins affect directly or indirectly the cellular levels of lipids involved in diverse signaling cascades, including sphingosine-1-phosphate and PI(4,5)P_2_. The mutual relations between flotillins and distinct lipids are key to the regulation of their involvement in numerous cellular processes.

## 1. Introduction

Flotillins are membrane-associated scaffolding proteins involved in numerous events which can be grouped in a handful of partially overlapping categories: intracellular signaling events, endocytosis, protein trafficking, actin cytoskeleton remodeling, and gene expression (Table 1). The interest in flotillins has been fueled additionally by the fact that they are overexpressed in a variety of cancers and are important regulators of oncogenic signaling and of cell invasiveness. We refer the reader to earlier reviews addressing this and several other aspects of the structure and functions of flotillins [1,2,3,4] and focus here on the critical input of lipids to the regulation of flotillin functioning. Flotillin-1 and -2, also known as reggie-2 and reggie-1, respectively, are ubiquitously expressed proteins of about 48 kDa. They were discovered by the Stuermer’s group as proteins upregulated in goldfish retinal ganglion cells during regeneration of axons, which earned them the reggie name [5]. Simultaneously, they were found among proteins floating in density gradients during fractionation of detergent cell lysates and named flotillins [6,7]. In this review, the latter terminology is used.

Flotillin-1 and -2 have the same domain architecture and show sequence identity reaching about 50% for the human proteins [58,59]. They comprise an SPFH domain starting several amino acids downstream of the N-terminus and encompassing about 180 amino acids and an adjacent flotillin domain similar in size. Several distinct lipid- and protein-binding motifs have also been identified in flotillins [59,60,61,62], as shown in Figure 1. Owing to the presence of the SPFH domain, flotillins belong to the SPFH superfamily of proteins named after stomatin, prohibitin, flotillin, and HflC/K. Structurally related proteins with moderate sequence homology to human flotillins are found in bacteria, plants, and fungi, suggesting evolutionarily conserved functions [59,63,64,65].

The SPFH and flotillin domains mediate, respectively, membrane binding and oligomerization of flotillins (Figure 1). Notably, flotillins also interact directly or indirectly with numerous proteins (Table 2). Thanks to these properties, flotillins can act as scaffolding proteins facilitating the assembly of submembrane multiprotein complexes crucial for various cellular processes. For this reason, factors affecting the binding of flotillins to membranes and their association with the plasma membrane nanodomains called rafts have been studied extensively. In addition, flotillin-1 seems to function also on its own, and we discuss here how the functioning of flotillins-1/2 and flotillin-1 alone is regulated. We concentrate on the binding of flotillins to membrane lipids and highlight the impact of the recent developments in the field of protein *S-*palmitoylation on the understanding of flotillin-mediated processes.

*S-*palmitoylation is a posttranslational modification of proteins catalyzed by 23 different palmitoyl acyltransferases from the zinc finger and Asp-His-His-Cys domain-containing (zDHHC) enzyme family and consists in the attachment of the saturated palmitoyl acyl chain (C16:0), delivered by palmitoyl-CoA, to a cysteine residue. The thioester bond formed can be cleaved by acylthioesterases, and this potentially reversible character of *S-*palmitoylation draws special attention when considering its consequences. *S-*palmitoylation affects intermolecular interactions, stability, and other posttranslational modifications of proteins. 

Notably, *S-*palmitoylated proteins gain a hydrophobic moiety which facilitates their association with membranes and often directs them to distinct plasma membrane domains, including rafts [73,74,75]. For these reasons, *S-*palmitoylation also determines the subcellular localization and functioning of flotillins. On the other hand, flotillins affect directly or indirectly cellular levels of signaling lipids, including sphingosine-1-phosphate (S1P) and phosphatidylinositols. These mutual relations between flotillins and distinct cellular lipids modulate multifarious cellular processes.

## 2. Acylation and Sphingosine Binding Affect Association of Flotillins with Membranes

Flotillins are almost exclusively membrane-bound and can localize to both the plasma membrane and endomembranes [17,26,35,60,69,76]. The membrane association of flotillins is determined by the acyl chain(s) attached and the interaction of protein hydrophobic regions (probably via hairpin insertion) with the cytosolic leaflet of membranes [26,60]. The membrane-binding regions have been mapped to the N-terminal part of flotillin-1 and -2 encompassing 35–40 amino acids [62]. Within these regions lie a hydrophobic stretch and the site(s) of acylation [59]; the involvement of the fatty acyl residue(s) in the membrane binding of flotillins has been confirmed experimentally, as described below. In addition, both flotillins are predicted to possess a second hydrophobic sequence contributing to the membrane binding, located in the middle part of the SPFH domain (amino acids 134–150/151 in human flotillins) [59], as shown in Figure 1. This hydrophobic stretch likely has a substantial input to the membrane association of flotillin-1, since a fragment of flotillin-1 encompassing amino acids 36–179 binds to membranes when expressed in cells while the corresponding fragment of flotillin-2 does not [62]. Accordingly, Langmuir monolayer studies have shown that the penetration of recombinant non-acylated flotillin-1 into a monolayer with a lipid composition mimicking the inner leaflet of the erythrocyte membrane exceeded that of flotillin-2 [77].

Recently, flotillin-1 and -2 were shown to bind a specific membrane lipid, sphingosine [57]. The binding was mapped to amino acid regions 3–185 and 8–188 of flotillin-1 and -2, respectively. These regions encompassed the whole SPFH domain and contained the two hydrophobic stretches mentioned above (Figure 1); several additional hydrophobic sequences were revealed in flotillin-1 and -2 sequences by the authors [57]. Thus, the hydrophobic interactions of flotillins with membranes correspond, at least to some extent, to the binding of sphingosine. Reciprocally, a knockout of flotillin-1 alone or of flotillin-1 and -2 in mice led to a depletion of sphingosine content in membranes of mouse embryonic fibroblasts (MEF) derived from these animals. The level of S1P, the phosphorylated derivative of sphingosine having signaling activity, was also reduced in the MEF cells and in several tissues of the knockout mice. Those data indicate that flotillins not only bind sphingosine but also affect its cellular level and thereby the amount of S1P. This feature should be taken into account when considering the metabolism of sphingolipids in cells [78,79]. Sphingosine is generated from ceramide synthesized de novo in the endoplasmic reticulum and is then phosphorylated locally to S1P by sphingosine kinase (SK) 2. The same kinase also catalyzes the phosphorylation of sphingosine in the nucleus and mitochondria. In contrast, in the plasma membrane, sphingosine is generated by the breakdown of sphingomyelin and is phosphorylated in situ to S1P by SK1. Thus, the recently discovered flotillin–sphingosine interaction seems crucial for both the S1P generation/signaling and for the dynamics of the flotillin binding to the plasma membrane and the endoplasmic reticulum, as discussed in the following sections.

Apart from the hydrophobic interactions with sphingosine, the second major determinant of the membrane binding of flotillins is their fatty acylation. Notably, the pattern of acylation is different for the two flotillins (Figure 1). Flotillin-1 is *S-*palmitoylated on Cys34 located within the first hydrophobic stretch of the protein [24,59,60]. The potential palmitoylation of Cys5 and Cys17 predicted by relevant algorithms has not been confirmed experimentally (https://swisspalm.org). Flotillin-2 is *N-*myristoylated on Gly2 and *S-*palmitoylated on three cysteine residues. Cys4 is the major *S*-palmitoylation site of flotillin-2, but *S-*palmitoylation site of Cys19 and Cys20, which are located in the first hydrophobic stretch, also takes place [26]. Thus, the pattern of flotillin-2 myristoylation/palmitoylation resembles that of the other well-characterized doubly acylated proteins, like Src family kinases. The *N-*myristoylation is mostly co-translational and follows the removal of the N-terminal methionine. The saturated myristic acyl chain (C14:0) from myristoyl-CoA is attached to—now N-terminal—Gly2. This *N-*myristoylation reaction is catalyzed by *N-*myristoyl transferase 1 or 2, and since no mammalian enzyme is able to split the amide bond formed between the glycine and the acyl chain, it is irreversible. In general, *N-*myristoylation allows a fairly weak binding of a protein to a membrane and facilitates subsequent *S-*palmitoylation of the protein on neighboring cysteine residue(s) by the integral membrane zDHHC enzymes, thereby governing a stronger membrane association of the protein [80]. This sequence of events most likely also determines the acylation of flotillin-2, as its Gly2Ala mutant is not *S-*palmitoylated [26]. The membrane association of flotillin-2 weakens progressively with mutations of Cys4, Cys20, and Cys19. Finally, the Gly2Ala, hence not acylated at all, form of flotillin-2 exhibits only a residual ability to associate with membranes [26] that can be ascribed to its oligomerization with endogenous membrane-bound flotillins, as discussed below.

In the case of flotillin-1, it seems likely that the hydrophobic interactions with the lipid bilayer mediated by the SPFH domain provide the initial membrane anchor allowing its subsequent *S-*palmitoylation, as found, e.g., for phosphatidylinositol 4-kinase IIα [81]. The combination of Cys34 *S-*palmitoylation with the association of the hydrophobic segment(s) of the polypeptide chain with the inner layer of the plasma membrane produces strong membrane binding. This is confirmed by the resistance of wild-type flotillin-1 to dissociation from the membrane fraction in high-pH sodium carbonate buffer, a behavior characteristic for integral membrane proteins [60].

Thus, flotillin-1 binds to the membrane via its hydrophobic sequence(s) and is then *S-*palmitoylated on the sole Cys34 while flotillin-2 is first *N*-myristoylated, which provides the membrane anchor possibly aided by two hydrophobic stretch(es) of the protein and eventually is complemented by *S*-palmitoylation at up to three Cys residues. The differences between flotillin-1 and flotillin-2 in their respective combinations of potentially reversible *S-*palmitoylation(s) with the irreversible *N*–myristoylation could determine their preferential association with particular cellular membranes and membrane domains. The role and potential dynamics of the sphingosine binding by flotillins await elucidation. Until now, only an *S-*palmitoylation-dependent plasma membrane localization of flotillin-1 has been confirmed. In several cell lines, overexpressed wild-type flotillin-1 localized to the plasma membrane while a non-palmitoylated Cys34Ala mutant displayed intracellular location [24,38,60], although contrary results have also been reported [69]. Interestingly, inhibition of protein *S*-palmitoylation induces a shift of endogenous flotillin-1 from the plasma membrane to intracellular vesicles [82].

Besides acylation, oligomerization is another factor affecting the membrane binding of flotillins. Flotillins exist as a mixture of monomers and homo- and/or hetero-tetramers, as indicated by chemical cross-linking and by sucrose density gradient fractionation of endogenous proteins [24,62]. Homo-oligomerization of flotillin-2 was also indicated by the yeast two-hybrid system [26]. The homo-oligomerization of flotillin-2 is governed by fragments of the flotillin domain rich in GluAla repeats and hence predicted to form a coiled-coil structure (CC1 and CC2 in Figure 1) [62]. A flotillin-2 construct lacking the C-terminal amino acids 278–428 bound to the plasma membrane but did not accumulate in submembrane caps formed by flotillin-1 in T cells [14]. That result indicates that the CC2 stretch of flotillin-2, truncated in the construct, is also required for hetero-oligomerization of flotillins (Figure 1).

The relation between the acylation of flotillins and their oligomerization is not clear. The Gly2Ala mutation of flotillin-2 mentioned earlier did not preclude its oligomerization despite fully abolishing the acylation [26]; also, the Cys34Ala mutant of flotillin-1 was able to homo-oligomerize [38]. In stark contrast, recent studies on cells depleted of zDHHC5 showed that the reduced *S-*palmitoylation of endogenous flotillin-2 correlated with an absence of its oligomers [83]. A plausible explanation of this discrepancy is that overexpressed tagged flotillin constructs tended to aggregate in cells [60]. The possibility that *S-*palmitoylation of flotillin-2 could facilitate its oligomerization indicates an intriguing mode of modulation of flotillin functioning. The ability of flotillins to form hetero-oligomers has led to the proposal that those oligomers could serve as scaffolds for other protein complexes involved in cell signaling, endocytosis, protein sorting, and cell migration [17,28,29,34,62,84,85]. By shifting the balance between flotillin monomers and oligomers toward the latter, *S-*palmitoylation of flotillins could modulate these processes. Moreover, co-localization of *S-*palmitoylated flotillin-1 and -2 within distinct plasma membrane domains, like rafts, can facilitate formation of functional hetero-tetramers and higher-order hetero-oligomers of flotillins. These processes can precede endocytosis involving flotillins, as suggested by Babuke et al. [76]. In addition, *S-*palmitoylation can facilitate interactions of flotillins with other S-palmitoylated proteins residing in rafts, like the scaffolding membrane palmitoylated protein 1 (MPP1) [82]. The study of Babuke et al. [76] has brought about an additional twist in the understanding of flotillin oligomerization by indicating that Tyr163 of flotillin-2 is involved in its hetero-oligomerization with flotillin-1 (but not homo-oligomerization). In further sections, we discuss how *S-*palmitoylation, hetero-oligomerization, tyrosine phosphorylation, and raft association of flotillins can control endocytosis mediated by these proteins.

It is worth mentioning that oligomerization can to some extent affect the membrane binding of flotillins. Thus, overexpressed deletion constructs of flotillin-1 and -2 containing only the flotillin domain (amino acids 180/184–428) or lacking N-terminal 35–40 amino acids and the non-acylated Gly2Ala form of flotillin-2 all maintained to some extent the ability to associate with membranes, most likely through oligomerization with endogenous flotillins [26,62]. This feature should be taken into account when designing and interpreting experiments; overexpressed flotillin constructs devoid of an intrinsic ability to bind membranes can act as dominant negative mutants by sequestering endogenous proteins, thereby revealing processes in which flotillins are involved. Indeed, ectopic expression of a C-terminal fragment of flotillin-2 (amino acids 184–390) or the Gly2Ala form of flotillin-2 interfered with cytoskeleton reorganization in activated T cells [14,29]. Furthermore, ectopic expression of the Cys34Ala non-palmitoylated mutant of flotillin-1 suppressed signaling of insulin-like growth factor-1 (IGF-1) receptor in HeLa cells and inhibited endocytosis of dopamine transporter (DAT) in HEK293-derived EM4 cells despite the presence of normal levels of endogenous flotillin-1 and -2 [24,38].

## 3. Flotillins Associate with Plasma Membrane Rafts

The discovery of flotillins coincided with a period of most extensive studies on heterogeneity of the plasma membrane organization. Those studies were triggered by the hypothesis of the existence of so-called membrane rafts put forward in late 1990s by the Simon’s group. Rafts were envisioned as sphingolipid- and cholesterol-rich (micro)domains that acquire liquid ordered phase and thereby separate from the disordered glycerophospholipid-enriched milieu in the plasma membrane and *trans-*Golgi membranes [86]. The sphingolipid/cholesterol-rich domains of the *trans*-Golgi network can be involved in the Golgi-to-plasma membrane transport of glycosylphosphatidylinositol (GPI)-anchored proteins [87]. It has been shown recently that submicromiter-sized sphingomyelin/cholesterol-rich microdomains in late phagosomes facilitate clustering of dynein at the membrane surface and thereby strengthen the interaction of this motor protein with a single microtubule for a directional movement of the vesicles toward lysosomes [51]. Notably, plasma membrane rafts have been indicated to serve as signaling platforms for distinct receptors, like T cell receptor (TCR) and other immunoreceptors, by causing a local enrichment of GPI-linked proteins anchored into the external leaflet of rafts and of signaling proteins, such as *N*-myristoylated and *S*-palmitoylated tyrosine kinases of the Src family, associated with the inner leaflet of the membrane. [88,89,90,91]. Although initially questioned and modified, the concept of an involvement of rafts in signal transduction is now widely accepted, having found direct support in microscopy studies made possible by recent achievements of bio-imaging techniques [92,93].

Flotillins have often been considered as markers of plasma membrane rafts based on ample biochemical studies involving fractionation of detergent cell lysates over sucrose or OptiPrep (iodixanol) density gradients, leading to separation of a lipid-rich protein-poor low-density fraction roughly corresponding to rafts and named detergent resistant membranes (DRM). Flotillins are among the few proteins which associate with the inner leaflet of the plasma membrane and are isolated in the DRM fraction. Other such proteins are caveolins, a cholesterol-binding proteins assembling within sphingolipid/cholesterol-rich regions of the plasma membrane into flask-shaped invaginations called caveolae. Since flotillins float in density gradients also in cells devoid of caveolae and separate from caveolae in intact cells, they have been proposed to provide a scaffold for the assembly of the flat (as opposed to caveolae) microdomains of the plasma membrane [6,26,35,82]. Another line of data supporting a relation of flotillins with rafts come from immunofluorescence and electron microscopy studies, indicating partial co-localization of flotillins and GPI-anchored proteins after their clustering [18,35,84,94]. Moreover, recent studies of the Sikorski’s group have indicated that flotillins interact with another *S*-palmitoylated raft-associated protein—MPP1 of the membrane-associated guanylate kinase (MAGUK) family of scaffolding proteins—and together they contribute to the assembly of rafts in the erythrocyte membrane [66,82].

Recent detailed fractionation studies have indicated, however, that flotillins can in fact separate from the “classical” low density DRM fraction and instead locate in DRM fractions of a higher density [24]. Such “heavy” flotillin-rich DRM fractions displayed lower sensitivity to cholesterol depletion than did the buoyant DRM, thus resembling the “heavy” protein-rich DRM described earlier by Horejsi’s group in relation to TCR signaling [95]. To sum up, flotillins organize at the inner leaflet of the plasma membrane in assemblies which share some but not all properties/localization with sphingolipid/cholesterol-rich rafts accommodating GPI-anchored protein in the outer leaflet of the membrane. Most likely, due to their *N-*myristoylation, *S-*palmitoylation and hetero-oligomerization flotillins are anchored in plasma membrane domains of a similar lipid composition to those surrounding the GPI-anchored proteins and the Src family tyrosine kinases. Notably, in unstimulated cells, these domains can be separated from each other and merge together after cell stimulation, as shown in Figure 2 for flotillin-mediated endocytosis.

The motif(s) mediating the raft localization of flotillins were identified taking advantage of the insensitivity of DRM to nonionic detergents. A contribution of the N-terminal 35 amino acids was established for fotillin-1, since deletion of this fragment increased markedly its Triton X-100 solubility [69]. Notably, mutation of Cys34 alone did not affect the detergent solubility of flotillin-1 [69]. Thus, *S-*palmitoylation can be crucial for the plasma membrane targeting of flotillin-1 (at least in a majority of cells tested, as discussed above), but to the best of our knowledge, its contribution to the raft association of flotillin-1 does not seem to be essential. In the case of flotillin-2, its buoyancy in detergent density gradients correlated with palmitoylation of Cys4 and Cys19, indicating that these modifications together with the *N-*myristoylation of Gly2 target flotillin-2 to rafts. Nevertheless, even the non-acylated Gly2Ala flotillin-2 mutant displayed residual association with DRM. On the other hand, the N-terminal 30-amino-acid fragment of flotillin-2 was soluble in Triton X-100. Taken together, the data indicate that *N*-myristoylation and *S*-palmitoylation of flotillin-2 facilitate but are not sufficient to direct it to the DRM fraction, and likely, also intermolecular interactions mediated by the SPFH and/or the flotillin domain are involved [26].

Finally, also the binding to sphingosine and cholesterol can affect the association of flotillins with rafts. The sphingosine binding is mediated by the SPFH domain of flotillins [57] but the exact identity of the hydrophobic sequences of flotillins engaged is not known. A knockout of flotillin-1 or flotillin-1/2 led to the depletion of sphingosine in the DRM fraction of MEF cells, indicating that flotillins not only bind to sphingosine but also affect its level in rafts. Taking into account that sphingosine is generated in the plasma membrane upon sphingomyelin breakdown, the sphingosine binding can affect the dynamics of the flotillin–raft association.

The binding of cholesterol by flotillins is mediated by the cholesterol recognition/interaction amino acid (CRAC) motif(s) located within the SPFH domain (amino acids 117–124 in flotillin-1; 120–127 and 157–169 in flotillin-2) [61]. Although cholesterol depletion does not affect the distribution of flotillins in the DRM fraction [24,82], flotillins are considered as regulators of cholesterol uptake and trafficking in cells [54,72]. Moreover, the interaction of flotillins with cholesterol in late phagosomes was found to promote the formation of the dynein-clustering microdomains mentioned earlier [51].

In summary, several factors ensure the association of flotillins with plasma membrane rafts. The participation of *S-*palmitoylation seems critical mainly for flotillin-2, while hydrophobic interactions of fragments of the polypeptide chain with the bilayer can concern both flotillins. It remains to be elucidated to what extent these hydrophobic interactions can be ascribed to sphingosine binding; also, cholesterol as a raft anchor of flotillins cannot be ruled out. Finally, intermolecular interactions, including hetero-oligomerization of flotillins and their binding to other *S-*palmitoylated plasma membrane raft proteins, could be involved.

## 4. Association of Flotillins with Rafts Facilitates Endocytosis and Recycling of Certain Proteins

Plasma membrane rafts of resting cells are now considered to be extremely labile nanometer-sized structures formed due to interactions of sphingolipids and cholesterol, with a contribution of cell-surface GPI-anchored proteins and *S-*palmitoylated transmembrane/peripheral proteins and their oligomerization. These dynamic nanodomains can merge together upon cell stimulation into more stable platforms, thereby causing local accumulation of distinct receptors, proteins engaged in their signaling pathways, and lipids involved in these signaling cascades, such as sphingolipids and phosphatidylinositols [82,88,96,97,98,99,100,101].

The data pointing to a dynamic nature and the nanoscale of membrane rafts often contrast with those on the cellular localization of flotillins. In a number of cell types, flotillins form submembrane patches up to 1 μm in diameter [33,34,35]. T lymphocytes and other nonadherent hematopoietic cells are an extreme example where flotillins cap one pole of the cell only [102]. Upon TCR stimulation, these preexisting submembrane flotillin-rich domains condense together with signaling proteins, including raft-residing LAT adaptor protein and Lck tyrosine kinase, into a so-called immunological synapse [14,15,102,103]. During cell migration, the preassembled flotillin structures of hematopoietic cells polarize to the rear end of the cell called the uropod [28,29,103,104]. Flotillins are also concentrated at sites of cell–cell contacts. They associate with N- and E-cadherins, providing a scaffold for their clusters in the plasma membrane, thereby stabilizing adherens junctions [68,105]. This scaffolding property of flotillins involves their interaction with GM1 sphingolipid- and cholesterol-rich domains in the plasma membrane on one hand and binding of cadherin complexes and the actin cytoskeleton on the other [105]. In breast cancer cells, flotillins stabilize ErbB2 in the plasma membrane sustaining signaling activity [106]. In erythrocytes, flotillins contribute to the organization of membrane rafts in co-operation with MPP1 protein [66,82,107]. Thus, it can be proposed that flotillins bound to the inner leaflet of the plasma membrane facilitate the assembly of plasma membrane microregions originating from rafts.

Having stressed the stabilizing nature of the submembrane flotillin patches for the lateral organization of the plasma membrane lipids and proteins in some cells, one should also mention that flotillins undergo budding and internalization from the plasma membrane. Early studies indicated that a joint expression of flotillin-1 and flotillin-2 induces plasma membrane invagination and budding off of flotillin-decorated vesicles. Those studies also identified flotillin hetero-oligomerization as a factor determining this process [76,84]. Next, tyrosine phosphorylation and *S-*palmitoylation were added to the list of factors affecting the flotillin-mediated endocytosis [8,34,36]. Before discussing the role of *S-*palmitoylation in the internalization, the problem of the cargo of the flotillin-containing vesicles should be addressed. The relation of flotillins to the plasma membrane rafts suggests that flotillin-mediated endocytosis could serve for internalization of GPI-anchored proteins. Indeed, some GPI-anchored proteins, like Thy-1 and CD59, and also the raft glycosphingolipid GM1, co-localize with flotillins at the plasma membrane and in endosomes, especially after their cross-linking on the cell surface; flotillins have been found to be important for the internalization of those molecules [18,33,35,37]. Notably, careful cell imaging studies have indicated that flotillin-decorated vesicles are separated from the clathrin-coated ones [33,84]. Silencing of flotillin-1 expression with siRNA reduced the endocytosis of CD59 by about 50% only [33]. Those data together with some immunofluorescence and ultrastructural analyses [108] called into question the strict dependency of the internalization of GPI-anchored proteins on flotillins. Other studies either confirmed or argued against an involvement of flotillins in the endocytosis of proteins internalized in clathrin-coated vesicles, like amyloid precursor protein (APP), DAT, and epidermal growth factor (EGF) receptor [1,9,38,39,42,109,110]. Finally, flotillins have been shown to mediate fluid phase uptake [33], and recent studies on this process shed new light on the flotillin-mediated internalization of plasma membrane proteins and its relation to rafts and protein S-palmitoylation [34], as discussed in the next section.

Simultaneously, a prominent line of studies which benefited from the development of bioimaging techniques have indicated that flotillins participate in the recycling of certain plasma membrane receptors and proteins between the plasma membrane and Rab11-positive recycling endosomes. The recycled proteins include TCR, α-amino-3-hydroxy-5-methyl-4-isoxazolepropionic acid (AMPA) receptor, DAT, E-cadherin, and integrin α-5/β-1 [16,17,38,39,45,46,47]. Their recycling is required, respectively, for sustained TCR and AMPA receptor signaling, dopamine uptake at the presynaptic membrane, and assembly/turnover of adherens junctions and focal adhesions. In cancer cells, a transmembrane metalloproteinase MT1-MMP can traffic with flotillin assistance from the plasma membrane to atypical nondegradative Rab7-positive endosomes. The following exocytosis of the enzyme from invadopodia leads to a local enrichment of MT1-MMP, facilitating degradation of the extracellular matrix by metastatic cells [48]. The involvement of flotillins in TCR recycling corresponds with their association with the plasma membrane rafts. Thus, the flotillin-based microdomains are proposed to guide TCR associated with these domains from the plasma membrane toward Rab5- and next Rab11-positive recycling endosomes. However, neither the endocytosis of TCR from the plasma membrane itself nor the return of the receptor from the Rab11-marked endosomes to the plasma membrane require flotillin assistance. Instead, flotillin participation is confined to “pre-early endosomal sorting” of TCR, which coincides with its internalization from the plasma membrane [17]. As TCR is an iconic receptor which relies on rafts for signal transduction [90], the coalescing of rafts promoted by flotillins can govern the entry of TCR into the recycling route.

Studies on TCR recycling have revealed that flotillins not only control the sorting of TCR for its subsequent recycling but also determine the spatial organization of the Rab5- and Rab11-positive endosomes. Redpath and coworkers [17] suggested that this flotillin function resembles that in late phagosomes [51]. On one hand, flotillins interact directly or indirectly with proteins driving the trafficking of the vesicles along microtubules, including SNX4, Rab11, and dynein [46], and on the other, they are prone to associating with sphingomyelin/cholesterol-rich domains or even can promote the assembly of such domains in the endosome membrane. Clustering of dynein follows, driving rapid transport of the vesicle along microtubules [51]. One is tempted to speculate that the plasma membrane-derived raft-originating domains in endosomes stabilized by flotillins govern the trafficking of TCR through the Rab5- and Rab11-positive compartments. It remains to be established whether similar lipid-dependent interactions of flotillin with endosomes contribute also to the flotillin assistance in the recycling of other cell surface proteins [39,45,46,47].

In conclusion, flotillin-based microdomains can serve as platforms mediating formation of multiprotein complexes at the plasma membrane and recycling of distinct cell surface proteins; their involvement in endocytosis of some cargo proteins is also conceivable. Obviously, multiple factors regulate those processes among which the propensity of flotillins to associate with sphingolipid/cholesterol-rich rafts of the plasma membrane and corresponding domains in endomembranes seems to play a major role.

A recent study on the uptake of a fluid phase marker, dextran, by human retinal pigment epithelial cells has refreshed our view of the involvement of flotillins in endocytosis. In that study, fluid phase internalization was intensified upon treatment of cells with ultrasound, microbubbles, and desipramine (USMB), and this treatment could arguably be applied also for drug delivery to cancer cells. The uptake required the participation of flotillin-1 and -2, zDHHC5, and Fyn tyrosine kinase of the Src family [34], suggesting that S-palmitoylation of flotillins by zDHHC5 underlies the relation between rafts and flotillin-mediated endocytosis and possibly also other flotillin functions, as discussed below.

## 5. Flotillin-Mediated Endocytosis Requires Protein S-palmitoylation

zDHHC5 was unequivocally identified as the palmitoyl acyltransferase catalyzing S-palmitoylation of flotillin-2 by studies on zDHHC5-deficient mice. The zDHHC5 deficiency abolished S-palmitoylation of flotillin-2 in neural stem cells and decreased it by about 50% in whole brain lysates [83]; possible participation of zDHHC5 in the S-palmitoylation of flotillin-1 can also be inferred from those data. Subsequent studies on COS and HEK293-dervied cells overexpressing or depleted of zDHHC5 confirmed its involvement in flotillin-2 S-palmitoylation at all three cysteine residues [83,111].

zDHHC5 is one of the few palmitoyl S-acyltransferases localized to the plasma membrane and endosomal compartments, while most members of this family reside in the endoplasmic reticulum and Golgi apparatus [74]. Studies on the localization of zDHHC5 and its substrates suggest that flotillin-2 is S-palmitoylated at the plasma membrane [111,112,113]. zDHHC5 also mediates S-palmitoylation of cell surface proteins required for so-called massive endocytosis (MEND) thought to originate from ordered regions of the plasma membrane (rafts?) [114]. An intriguing relation of MEND to endocytosis involving flotillins has been suggested [34] and is discussed below. On the other hand, zDHHC5 can also localize intracellularly to sorting and recycling endosomes, possibly overlapping with the distribution of flotillins. Furthermore, zDHHC5 has been shown to affect several aspects of the Golgi–plasma membrane–endosome trafficking of proteins [115,116]. The most comprehensive picture of the zDHHC5 localization, its substrates, and functions concerns neurons. In these cells, zDHHC5 is located in the postsynaptic membrane, where it forms a ternary complex with the tyrosine kinase Fyn and PSD-95 scaffolding protein. This complex regulates the synaptic turnover of AMPA-type glutamate receptors, and details of its assembly have been revealed pointing to the importance of the interactions between the PDZ3 domain of PSD-95 and its binding motifs present in the two other proteins [117]. Those authors postulate that phosphorylation of zDHHC5 by the Fyn kinase is required for keeping zDHHC5 in the postsynaptic membrane. Upon neuron activation, depalmitoylation of PSD-95 and the correlated inhibition/displacement of Fyn from the plasma membrane lead to zDHHC5 dephosphorylation, clathrin-dependent internalization and targeting to recycling endosomes. There, zDHHC5 S-palmitoylates δ-catenin and these two proteins together return to the postsynaptic membrane, where δ-catenin interacts with N-cadherin, strengthens the synapse structure, and stabilizes AMPA receptors [117]. Even though flotillins are abundant in neurons and have been shown to participate in neuronal protein recycling through Rab11-positive endosomes [47] and in the formation of glutamatergic synapses in hippocampal neurons [118,119], no studies have so far been devoted to the details of flotillin-2 (and possibly flotillin-1) S-palmitoylation by zDHHC5 in neurons.

Flotillin-2 does not co-immunoprecipitate with zDHHC5, yet its S-palmitoylation requires an involvement of an unstructured fragment of the C-terminal part of zDHHC5 [111]. However, flotillin-2 carries a PDZ3-binding motif [59] (Figure 1) possibly involved in an interaction with a putative adaptor protein linking flotillin-2 with zDHHC5. The most likely candidates for this function are adaptor proteins of the MAGUK family, to which PSD-95 and MPP1 belong, as suggested by studies of the Sikorski’s group [66,82]. Notably, flotillins are phosphorylated by Fyn kinase, flotillin-1 at Tyr160, and flotillin-2 at Tyr163, and this phosphorylation is required for flotillin-dependent endocytosis [8,36]. Flotillin-1 also associates with the Lyn tyrosine kinase, another member of the Src family [20]. It should be mentioned that data arguing against a role of tyrosine phosphorylation of flotillins in the endocytosis have also been reported. Instead, an involvement of Tyr163 of flotillin-2 in its hetero-oligomerization with flotillin-1 was indicated [76].

The already mentioned recent study on endocytosis triggered by the USMB treatment of human retinal pigment epithelial cells provided a compelling data on the operation of the zDHHC5–Fyn–flotillin triad at the plasma membrane [34]. The treatment induced massive fluid phase endocytosis that depended on the formation of flotillin-positive vesicles and was inhibited by joint silencing of flotillin-1 and -2 or silencing of zDHHC5 or Fyn kinase. The USMB treatment led to an enrichment of zDHHC5 in the plasma membrane. In contrast to neurons, however, the plasma membrane localization of zDHHC5 in the retinal pigment epithelial cells was facilitated by its dephosphorylation, which, oddly enough, required a contribution from Fyn. Flotillin phosphorylation by Fyn was not addressed in that study [34].

On the basis of the data discussed above, the following mechanisms controlling flotillin-mediated endocytosis can be considered: upon a stimulus, like USMB, Fyn kinase catalyzes phosphorylation of flotillins and affects (directly or indirectly) phosphorylation of zDHHC5 promoting its plasma membrane localization. There, zDHHC5 catalyzes S-palmitoylation of flotillins and other cell surface proteins. S-palmitoylation of flotillins (mainly flotillin-2) triggers the assembly of submembrane flotillin-rich complexes with flotillin hetero-tetramers being their predominant building blocks. Formation of these complexes is likely to involve plasma membrane rafts for the reasons presented in the previous sections, including protein S-palmitoylation. Namely, S-palmitoylation of flotillin-2 at up to three cysteine residues can enhance its association with rafts [26]. This association can also be increased by a concomitant S-palmitoylation of other cell surface proteins by zDHHC5 shifting the balance of the membrane lipid phase toward a more ordered one and thereby facilitating coalescence of the ordered membrane domains [120,121]. Interactions of flotillin-2 with other S-palmitoylated proteins, especially those of the MAGUK family (e.g., MPP1), can also be involved. On the other hand, the binding of flotillins to sphingosine allows modulation of the flotillin–raft association due to the dynamics of sphingosine formation/phosphorylation in the plasma membrane [78,79]. Assuming that S-palmitoylation determines the association of flotillin-1 with the plasma membrane as well [24,60], the functional flotillin hetero-oligomers can form at the plasma membrane but not at endomembranes. The combination of all these factors determines the complex flotillin–zDHHC5–MAGUK proteins–Fyn kinase interactions, protein phosphorylation and S-palmitoylation, and the following coalescence of rafts and invagination of the membrane facilitated by flotillin hetero-oligomers. Endocytosis of the entrapped fluid phase and plasma membrane proteins comes next, the preferred cargo of the flotillin-demarked vesicles being GPI-anchored proteins due to their raft localization (Figure 2).

Fekri et al. [34] have pointed out that the USBM-induced endocytosis shares similarities with “lipid driven” MEND described by Hilgemann and coworkers [114]. Indeed, MEND also requires tyrosine phosphorylation of proteins and zDHHC5/2-dependent palmitoylation of cell surface proteins. All the factors that induce MEND, like high [Ca+2] increase, distortion of the outer leaflet of the plasma membrane by cleavage of outer-leaflet sphingomyelin to ceramide, enrichment of cells with cholesterol, and mild detergent treatment, are proposed to lead eventually to phase separation of ordered lipids within the plasma membrane, its invagination, and pinching off of endosomes. S-palmitoylation of proteins is likely the ultimate impulse for MEND, as the enrichment of the inner leaflet of the plasma membrane in the saturated acyl chains of palmitate can induce growth or coalescence of ordered domains (rafts?). Thus, S-palmitoylation of flotillins can be crucial for MEND and other forms of clathrin-independent endocytosis, which, according to Hilgemann and coworkers, represent a “continuum of related events varying in actin-dependence and membrane phase dependence” [114].

Interestingly, S-palmitoylation of proteins is also required for the flotillin-dependent internalization of α-synuclein, which leads to a concomitant endocytosis of DAT from the presynaptic membrane of dopaminergic neurons and is related to the formation of Lewy bodies and the development of Parkinson’s disease [39]. That recent study has revived the idea that flotillins can govern clustering of plasma membrane proteins, which are then internalized mainly via clathrin-coated vesicles, such as APP, DAT, and EGF receptor. Their flotillin-mediated clustering in the plasma membrane is suggested to precondition those proteins for the subsequent endocytosis [9,42,109]. Notably, this scenario resembles that believed to determine the flotillin-dependent entering of the recycling route by TCR [17], described in an earlier section. It can be hypothesized that all those flotillin-dependent processes can result from the global rearrangement of the plasma membrane organization caused by changes in the S-palmitoylation status of flotillins and other cell-surface proteins. These, in turn, can be induced by certain stimuli other than USMB, like clustering of GPI-anchored proteins, α-synuclein binding, and stimulation of cells with EGF. In addition, the flotillin-assisted clustering and/or internalization of DAT can also be regulated by serine phosphorylation of flotillin-1 by protein kinase C (PKC) [38].

The data discussed above indicate that flotillins function as components of multiprotein complexes comprising also zDHHC5, Fyn, and MAGUK proteins. This, in turn, suggests that a combination of S-palmitoylation and phosphorylation of those proteins can modulate the large variety of events involving flotillins (or assisted by flotillins, according to Meister and Tikkanen [2]), like stabilization of receptor complexes in the plasma membrane, endocytosis, and protein/receptor recycling.

Finally, having presented the flotillin involvement in intracellular processes, the presence/role of flotillins in extracellular vesicles (EVs) should also be addressed (Table 1). EVs are formed either by outward budding of the plasma membrane or by exocytosis of multivesicular bodies, the latter called exosomes. Therefore, although in general all EVs comprise a lipid bilayer with integral and peripheral membrane proteins and contain entrapped proteins and RNA, they are heterogeneous in size and composition [122]. Notably, flotillin-1 was found in all types of EVs released by human dendritic cells [123]. It remains unknown whether the localization of flotillins to EVs is affected by their fatty acylation. EVs are thought to be involved in intercellular communication and can thereby contribute to the pathogenesis of tumors, inflammation, and neurodegenerative diseases; consequently, EV components can serve as biomarkers of those pathological processes [122,124]. Among others, flotillin-2 in combination with some other exosomal proteins has been identified by mass spectrometry and immunoblotting analysis of urinary samples as a prostate cancer biomarker of potential diagnostic value [125]. On the other hand, Abdullah et al. [126] showed that the flotillin-1 level in serum and cerebrospinal fluid is significantly decreased in patients with Alzheimer’s disease as a result of diminished exosome release, and therefore, flotillin-1 can be used as a blood marker for the differential diagnosis of the disease from other cognitive impairments. Besides being promising biomarkers, flotillins can also affect exosome composition. Thus, owing to its CRAC motifs, flotillin-2 contributes to the recruitment of cholesterol to multivesicular bodies and the following exosomal secretion of cholesterol [54]. An involvement of flotillin-1 and -2 in modulating the content of caveolin-1 and annexin-2 in exosomes has also been indicated [55]. Notably, other studies suggest that changes of the exosomal content of caveolin-1 can follow changes of its synthesis at the endoplasmic reticulum that is affected by flotillin-1 [13], as described below.

## 6. Some Unique Functions of Flotillin-1 Are Determined by Its S-palmitoylation

In view of the different patterns of acylation of flotilllin-1 and flotillin-2, one can ask whether both flotillins always cooperate or whether they could also have some unique functions. Studies on this problem are hindered by the fact that the two proteins coexist in cells and protect each other from proteasomal degradation. Multiple studies point to a stabilizing effect of flotillin-2 on flotillin-1 [23,46,62,76,127,128]. Some data, like those on flotillin-2 expression in flotillin-1 knockout mice, indicate that also the stability of flotillin-2 depends to some extent on the presence of flotillin-1 [24,28,110]. It can be speculated that only a profound depletion of flotillin-1 affects flotillin-2 stability while the reciprocal effect is easier to achieve. This, in turn, could reflect differences in the expression of genes encoding flotillin-1 and flotilllin-2. We found that, in three different types of macrophage/macrophage-like cells, the level of flotillin-2 mRNA exceeds that of flotillin-1 (unpublished data), suggesting that a pool of cellular flotillin-2 exists apart from its hetero-oligomers with flotillin-1. If indeed the abundance of flotillin-1 is lower than that of flotillin-2, flotillin-1 could be limiting the assembly of the flotillin1/2 hetero-oligomers at the plasma membrane of macrophages.

A line of recent data indicates that flotillin-1 can also function independently of flotillin-2 and that some of its activities are regulated by S-palmitoylation of the single Cys34 of flotillin-1. They include the contribution of S-palmitoylated flotillin-1 to the transport of IGF-1 receptor from the endoplasmic reticulum toward the plasma membrane [24] and the translocation of non-palmitoylated flotillin-1 to the nucleus for upregulation of the Snail transcription factor [56] (see further). Thus, a knockdown of flotillin-1 in HEK293 and HeLa cells precluded the localization of IGF-1 receptor to the plasma membrane and thereby inhibited IGF-1-induced signaling and cell proliferation. All those deficits were reverted by ectopic expression of wild-type flotillin-1 but not its Cys34Ala non-palmitoylated mutant. Notably, the depletion of flotillin-1 did not lead to changes in the total cellular amount of IGF-1 receptor, which indicated that flotillin-1 affects transport of the receptor from the endoplasmic reticulum to the plasma membrane rather than its synthesis or degradation. The contribution of flotillin-2 to IGF-1 receptor trafficking was also addressed in that study, although it was not straightforward as the knockdown of flotillin-1 led to a depletion of flotillin-2, which could not be reversed by expression of Cys34Ala flotillin-1. However, flotillin-2 returned to a normal level after the activity of proteasomes was blocked; even though the flotillin-2 localized to the plasma membrane, this was not accompanied by a concomitant plasma membrane recruitment of IGF-1 receptor [24]. Those results suggest that S-palmitoylation of flotillin-1 in the endoplasmic reticulum is crucial for proper trafficking of flotillin-1 and IGF-1 receptor to the plasma membrane (Figure 3A), corroborating earlier results on the plasma membrane localization of S-palmitoylated flotillin-1 [60]. Activation of IGF-1 receptor in the plasma membrane occurs in domains enriched in flotillin-1 and flotillin-2 [24] (Figure 3A), in agreement with the results discussed in previous sections. Moreover, a prolonged IGF-1 receptor signaling requires de- and re-palmitoylation of flotillin-1. Taken together, the data suggest that flotillin-1 is S-palmitoylated by an unidentified zDHHC localized to the endoplasmic reticulum (zDHHCx in Figure 3A) but that its S-palmitoylation at the plasma membrane and/or endosomes during activation of IGF-1 receptor is also plausible. In the latter case, an involvement of zDHHC5 can be considered [83].

In the endoplasmic reticulum, non-palmitoylated flotillin-1 can be modified by the attachment of small ubiquitin-like modifier (SUMO)-2/3 at Lys51 and Lys195 by the E2 conjugating enzyme UBC9. No corresponding lysine residues are present in flotillin-2. The sumoylation of non-palmitoylated flotilllin-1 is typical for metastatic prostate cancer cells exposed to a mitogen. Sumoylated flotillin-1 translocates to the nucleus, where it interacts with the transcription factor Snail, thereby inhibiting its proteasomal degradation. It is tempting to speculate that the flotillin-1 translocation between the endoplasmic reticulum and the nucleus is linked with the sphingosine–S1P conversions in the respective membranes. The stabilized and therefore upregulated Snail induces expression of genes encoding proteins crucial for the endothelial-to-mesenchymal (ETM) transition and metastasis is induced (Figure 3B). Combined with the results indicating the importance of flotillin-1 palmitoylation for sustained IGF-1 signaling, these data suggest that the palmitoylation status of flotillin-1 at the endoplasmic reticulum and the plasma membrane affects proliferation and metastatic potential of prostate cancer cells [56].

A contribution of flotillin-1 to proper functioning of the endoplasmic reticulum has also been indicated by studies of Fork et al. [13] yet without addressing the role of its S-palmitoylation. Those authors found that a downregulation of flotillin-1 but not of flotillin-2 with shRNA led in human umbilical vein endothelial cells to a reduction of the level of several other proteins, including caveolin-1. Depletion of caveolin-1, in turn, reduced the scavenger receptor A-mediated endocytosis of a ligand of the Toll-like receptor (TLR) 3 triggering pro-inflammatory signals. Taken together, the depletion of flotillin-1 diminished the activation of this endosomal receptor. The drop of the caveolin-1 level was ascribed to endoplasmic reticulum stress caused by the flotillin-1 depletion and the following inhibition of the caveolin-1 synthesis de novo [13], i.e., a mechanism different from the flotillin-1-dependent trafficking of the IGF-1 receptor to the plasma membrane. Alternatively, a disabled exit of caveolin-1 from the endoplasmic reticulum in the absence of flotillin-1 could be the reason for the endoplasmic reticulum stress causing the overall downregulation of caveolin-1 [13] (Figure 3A). Thus, while the involvement of flotillin-1 in the functioning of the endoplasmic reticulum is firmly established, its mechanism(s) requires clarification. It is worth mentioning that activation of TLR4 with bacterial lipopolysaccharide (LPS), a strong pro-inflammatory agent, increases the level of palmitoylated flotillin-1 but not flotillin-2 in Raw264 macrophage-like cells, suggesting that S-palmitoylation of flotillin-1 could be important for the signaling activity of TLR4 [129].

To complete the picture of the unique properties of flotillin-1, one should mention its phosphorylation on Ser315 catalyzed by PKC and dephosphorylation by protein phosphatase 2A (PP2A) [38,130]. Notably, no homologous serine residue is present in flotillin-2. Several other serine residues of flotillin-1 and -2 are predicted by large-scale phosphoproteomic studies to undergo phosphorylation [131,132], but the physiological role of these modifications has not been studied. Flotillin-1 phosphorylated on Ser315 dissociates from the plasma membrane and apparently localizes to the cytosol [38,130]. In fact, in pulmonary artery epithelial cells, the majority of flotillin-1 was cytosolic [130], contradicting the established image of flotillins as (exclusively) membrane-bound proteins. Of note, the cytosolic flotillin-1 was not palmitoylated [38]. Hypothetically, the shift of flotillin-1 to the cytosol could also be linked with reduction of the sphingosine level in the plasma membrane (see next section), which together with the depalmitoylation would deprive flotillin-1 of its membrane anchors (Figure 3C). In pulmonary artery endothelial cells, only the membrane-bound flotillin-1 enhances cell migration and formation of endothelial barrier [130], most likely after assembling in hetero-oligomers with flotillin-2 at the plasma membrane (Figure 3C). One should be reminded here that Ser315 phosphorylation of fotilllin-1 by PKC also affects clustering and/or internalization of DAT [38,109], unlike the endocytosis of other cargo requiring tyrosine phosphorylation of flotillins (see Figure 2).

Finally, flotillin-1 controls yet another aspect of endocytosis, i.e., the sorting of ubiquitinated cell surface proteins, like EGF receptor, for degradation in lysosomes [12,49]. Flotillin-1 binds to the Hrs component of the endosomal sorting complex required for transpor (ESCRT)-0, allowing transfer of the ubiquitinated proteins onto the ESCRT-1 complex which determines their following degradation [12]. This function extends the list of protein trafficking processes that involve flotillins (Table 1). Whether any of the possible posttranslational modifications of flotillin-1 affects this aspect of its functioning has not been established yet.

In summary, the unique properties of flotillin-1, such as serine phosphorylation and S-palmitoylation on a single cysteine, determine its participation in diverse cellular process either alone or as part of the scaffolding flotillin-1/2 hetero-oligomers.

## 7. Flotillins Affect Sphingosine-1-phosphate Signaling and Are Indirectly Linked with PI(4,5)P_2_ Turnover

The functioning of flotillins is strongly dependent on lipids. Above, we have highlighted the leading role of the *S*-palmitoylation of flotillins, but one must not underestimate also the fact that the flotillin activity cross-sects with lipids key to intracellular signaling, mainly sphingolipids, and indirectly also phosphatidylinositols. The discovery of the interaction between flotillins and sphingosine led to the suggestion that flotillins could stabilize sphingosine in cellular membranes. This would allow its subsequent phosphorylation to S1P by sphingosine kinases [57]. Taking into account the signaling role of S1P, some of the flotillin activities could in fact be related to modulation of the S1P-dependent signaling. Quite unexpectedly, this mode of action has been found to involve regulation of gene expression: flotillin-1 knockout and concomitant S1P depletion reduced acylation of histones and thereby diminished the expression of selected genes, including encoding interferon-stimulated protein 15 [57].

Given the involvement of flotillins in endocytosis, it is of interest that the conversion of sphingosine to S1P by SK1 is crucial for endosomal membrane trafficking [133,134]. In fact, the hypothetical regulation of the flotillin–membrane association by the sphingomyelin–ceramide–sphingosine–S1P chain of conversions during endocytosis could affect the assembly/disassembly of flotillin scaffolds in the endosomal compartment in an analogous manner to the regulation of the assembly/disassembly of filamentous actin at forming and maturating phagosomes that is regulated by generation of phosphatidylinositol 4,5-bisphosphate (PI(4,5)P_2_) and its following hydrolysis and phosphorylation [135]. Moreover, flotillin-1 functioning and SK1 activity could be regulated in a coordinated manner because both these phosphoproteins are subject to dephosphorylation by PPA2. Indeed, the dephosphorylation of flotillin-1 by PPA2 in endothelial cells has been found to strengthen the integrity of the endothelial barrier and an input of S1P signaling into this process was also postulated [130] (Figure 3C).

Flotillins have also been found to affect phosphatidylinositol signaling, albeit indirectly. In T cells stimulated for migration with the SDF-2 chemokine, flotillins were found to be involved in targeting phosphatidylinositol 4-phosphate 5-kinase type Iγ90, one of the kinases catalyzing PI(4,5)P_2_ generation, to the uropod. Following its local generation, PI(4,5)P_2_ activates the ezrin/radixin/moesin proteins required for uropod retraction [30,31]. A co-localization of PI(4,5)P_2_ and flotillins was also observed in pathological neuronal structures typical for the brains of Alzheimer’s disease patients, suggesting their common involvement in neurodegeneration [136]. Flotillins have also been indicated as regulators of the PI3-kinase/AKT pathway crucial for cancer cell survival [137,138]. PI(4,5)P_2_ can be enriched in rafts, and a significant elevation of its level in rafts occurs during the LPS-induced clustering of CD14, a GPI-anchored plasma membrane raft protein assisting the TLR4 receptor of macrophages in pro-inflammatory signal transduction [99,139]. Given the raft association of flotillins, a link between their scaffolding property and the PI(4,5)P_2_ generation in rafts seems likely.

## 8. Conclusions

Since their identification over two decades ago, the structure and functions of flotillins have been studied extensively, producing robust data showing that flotillins are scaffolding proteins operating at the membrane–cytosol interface. This scaffolding property of flotillins is determined by their hetero-oligomerization and binding of actin and other proteins on one hand and their association with membranes on the other. The preferential association of flotillins with the plasma membrane rafts facilitates markedly their scaffolding potential since the distinct lipid composition of rafts drives local accumulation not only of flotillins but also of other select “raftophilic” proteins. Recent studies have revealed the details of how the dynamic nature of the flotillin-mediated processes is regulated, pointing to *S-*palmitoylation as an important factor. Thus, S-palmitoylation of flotillins, especially flotillin-2 acylated by zDHHC5 at multiple cysteine residues and concomitant S-palmitoylation of other cell surface proteins induced by specific stimuli like clustering of GPI-anchored proteins can lead to phase separation of ordered membrane domains (rafts) and their subsequent endocytosis. In a similar manner, protein sorting and recycling through endosomal compartments can be driven by lipids and assisted by flotillins. Flotillin-1 differs from flotillin-2 by not undergoing co-translational *N-*myristoylation and by being *S-*palmitoylated at a single cysteine only. Since the membrane binding of flotillins also involves interactions with sphingosine, the reduction of the sphingosine level in the plasma membrane or the endoplasmic reticulum could lead to dissociation of non-palmitoylated flotillin-1 from these membranes. Such membrane-detached flotillin-1 can translocate to the cytosol, as observed in endothelial cells or after sumoylation shift to the nucleus, and can upregulate gene expression via the Snail transcription factor. *S-*palmitoylation of flotillin-1 also determines the trafficking of IGF-1 receptor to the plasma membrane by a yet-to-be-characterized mechanism possibly related to the endoplasmic reticulum homeostasis. Assuming lower expression of flotillin-1 than flotillin-2, the supply of *S-*palmitoylated flotillin-1 can also be limiting the assembly of functional hetero-oligomers of flotillins at the plasma membrane during signal transduction by distinct receptors and flotillin-mediated endocytosis. Future studies are expected to provide more details on the *S-*pamitoylation of flotillins and to fully unveil the role of the flotillin–sphingosine–S1P interactions in the regulation of flotillin-mediated processes.

## Figures and Tables

**Figure 1 ijms-21-02283-f001:**
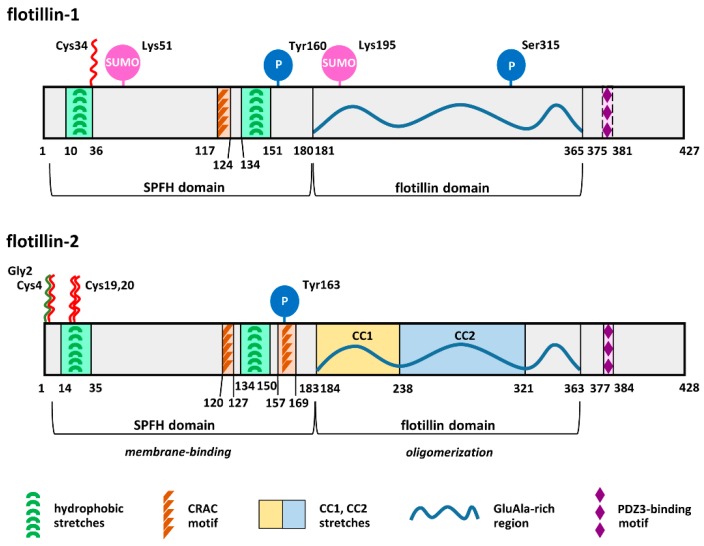
Structure of human flotillin-1 and flotillin-2: Amino acid sequences of human, mouse, and rat flotillin-1 are at least 97.9% identical, and those of flotillin-2 are at least 99.3% identical. The SPFH and flotillin domains are indicated according to References [59,60,62]. The PDZ3-binding domain is better conserved in flotillin-2 than in flotillin-1 [59]. Flotillin-1 and -2 possess, respectively, nine and eight potential sites of tyrosine phosphorylation [8]. Phosphorylation of Tyr160 and Tyr163 by Fyn kinase is confirmed [36]. *N*-myristoylation and *S*-palmitoylation are marked by green and red zigzags, respectively. PDZ domain, a domain present in PSD-95, Dlg, and ZO-1/2 proteins.

**Figure 2 ijms-21-02283-f002:**
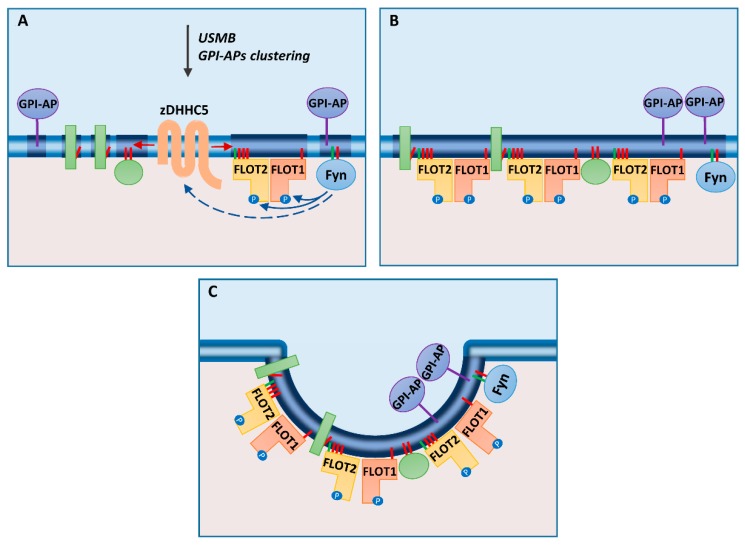
Flotillin-mediated endocytosis: (**A**) After stimulation of cells with ultrasound, microbubbles, and desipramine (USMB) or clustering of glycosylphosphatidylinositol (GPI)-anchored proteins (GPI-APs), zDHHC5 is activated indirectly by Fyn kinase, which also phosphorylates flotillins. (**B**) *S*-palmitoylation of flotillins and other cell-surface proteins induces coalescence of ordered regions (rafts) of the plasma membrane. (**C**) Flotillins can drive invagination of the membrane preceding pinching off of a vesicle. *N*-myristoylation and *S*-palmitoylation are marked by green and red bars, respectively. Transmembrane and peripheral membrane proteins are indicated by green rectangles and spheres, respectively.

**Figure 3 ijms-21-02283-f003:**
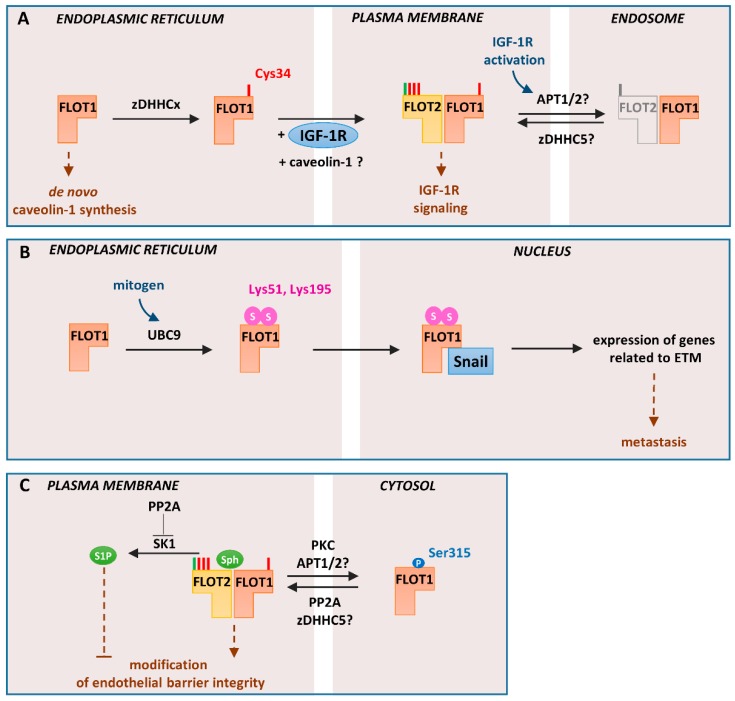
Cellular localization and functioning of flotillin-1 regulated by *S-*palmitoylation, sumoylation, and serine phosphorylation: (**A**). Trafficking from the endoplasmic reticulum to the plasma membrane. Newly synthesized flotillin-1 is *S-*palmitoylated by unidentified zDHHC (zDHHCx). This acylation determines trafficking of flotillin-1 to the plasma membrane concomitant with IGF-1 receptor (IGF-1R) trafficking. The exit of *S-*palmitoylated flotillin-1 can also mediate efflux of caveolin-1 from the endoplasmic reticulum protecting the endoplasmic reticulum from stress, which otherwise would inhibit the synthesis of caveolin-1. At the plasma membrane, flotillin-1 forms hetero-oligomers with flotillin-2 and, upon activation of IGF-1R, undergoes depalmitoylation/repalmitoylation required for prolonged receptor signaling. zDHHC5 and protein depalmitoylases—acyl-protein thioesterase-1/2 (APT1/2) or ABHD17 proteins—probably catalyze these reactions. Since the *S-*palmitoylation of flotillin-1 is required for its plasma membrane localization, depalmitoylation can be linked with its cycling between the plasma membrane and endosomes either alone or with flotillin-2. (**B**) Regulation of gene expression in metastatic prostate cancer cells: In these cells, non-palmitoylated flotillin-1 in the endoplasmic reticulum undergoes sumoylation at Lys51 and Lys159 with SUMO-2/3. The reaction is catalyzed by the E2 ligase UBC9. Sumoylated flotillin-1 translocates to the nucleus, binds Snail transcription factor, and protects it from proteasomal degradation. This triggers expression of genes related to ETM transition and metastasis. (**C**) Endothelial barrier regulation: In endothelial cells, flotillin-1 can be phosphorylated at Ser315 by protein kinase C (PKC). The phosphorylated and probably depalmitoylated flotillin-1 localizes to the cytosol. Subsequent dephosphorylation of flotillin-1 by PP2A allows its plasma membrane association, likely correlated with *S-*palmitoylation catalyzed by zDHHC5. PP2A-dephosphorylated flotillin-1 contributes to endothelial barrier integrity and angiogenesis. PP2A also inhibits the activity of SK1 which catalyzes phosphorylation of sphingosine, a lipid bound by flotillins. P, phosphorylation; S, sumoylation; Sph, sphingosine. *N*-myristoylation and *S*-palmitoylation are marked by green and red bars, respectively.

**Table 1 ijms-21-02283-t001:** Cellular processes involving flotillins.

	**SIGNALING**	
**Receptor**	**Affected events**	**Ref.**
EGF receptor	signaling leading to cell adhesion	[8]
	EGF receptor clustering and phosphorylation, ERK1/2 and Akt phosphorylation; flotillins as MAP kinase scaffolding proteins	[9]
	EGF receptor expression in breast cancer cells	[10]
	activation of H-Ras in breast cancer cells	[11]
	EGF receptor sorting and lysosomal degradation	[12]
TLR3	ligand internalization	[13]
TCR	raft association and recycling	[14,15,16,17]
PrP	[Ca^+2^] increase, Fyn and ERK1/2 activation, N-cadherin trafficking	[18,19]
IgE receptor	[Ca^+2^] increase, phosphorylation of IgE receptor γ chain and ERK1/2	[20]
insulin receptor	insulin-induced glucose uptake via Glut4 transporter	[21]
Gq protein- coupled receptors	p38 phosphorylation	[22]
not determined	signaling leading to axon regeneration: Rho GTPase activation, formation of N-WASP-Arp3/cortactin complexes, p38, ERK1/2 and FAK kinase phosphorylation	[23]
IGF-1 receptor*	IGF-1 receptor transport from endoplasmic reticulum to plasma membrane	[24]
integrins	ERK2 and FAK kinase phosphorylation	[25]
	**ACTIN CYTOSKELETON REMODELING**	
	**Affected processes**	
	filopodia formation	[26]
	cell spreading	[8]
	proper localization of Vav during T cell spreading	[14]
	F-actin binding	[27]
	axon regeneration	[23]
	uropod formation in neutrophils	[28]
	uropod formation in T cells, activation of ezrin/radix/moesin, localization of PIP5KIγ to uropod	[29,30,31]
	formation of lamellipodia at the growth cone of neurons	[32]
	cell adhesion and migration via α-actinin binding	[25]
	**ENDOCYTOSIS**	
	**Cargo**	
	fluid phase (magnetic dextran, AF488-dextran)	[33,34]
	GPI-anchored proteins	[18,33,35,36,37]
	DAT	[38,39]
	proteoglycans, e.g., syndecan-1	[40,41]
	APP	[42]
	semaphorin 3A receptor	[43]
	leucine-rich amelogenin peptide	[44]
	**PROTEIN TRAFFICKING**	
	**Protein/receptor**	
	**recycling**	
	DAT	[39]
	TCR	[16,17]
	α-5 and β-1 integrin	[45]
	E-cadherin	[46]
	AMPA receptor (GluA1, GluN1 subunits)	[47]
	MT1-MMP**	[48]
	**trafficking to lysosomes**	
	EGF receptor	[12]
	BACE1	[49]
	pseudokinase MLKL	[50]
	proteoglycans and bound ligands, including VLDL	[40,41]
	late phagosomes	[51]
	**from endoplasmic reticulum to plasma membrane**	
	IGF-1 receptor*	[24]
	caveolin-1*	[13]
	**retrograde transport**	
	cholera toxin	[52]
	Shiga toxin	[53]
	**RELEASE OF EXTRACELLULAR VESICLES**	
	**Affected processes**	
	regulation of cholesterol content in exosomes	[54]
	regulation of caveolin-1 and annexin-2 content in exosomes***	[55]
	**GENE EXPRESSION**	
	**Gene**	
	encoding proteins of ETM transition*	[56]
	encoding interferon-stimulated protein 5*	[57]

*dependent on flotillin-1; **via nondegradative Rab7-positive endosomes; ***an influence of flotillin-1 on de novo synthesis of caveolin-1 possible [13].

**Table 2 ijms-21-02283-t002:** Flotillins and their protein partners.

Protein	Method Used for Identification of the Binding to Flotillin-1 and/or Flotillin-2	Ref.
F-actin	flotillin-2 (SPFH domain), *direct binding*, in vitro binding assay	[27]
MPP1	flotilllin-1 and -2, *direct binding*, overlay assay	[66]
Lyn	flotillin-1, *direct binding*, mammalian two-hybrid system	[20]
	flotillin-1, co-immunoprecipitation	
Gαq	flotillin-1 and -2, *direct binding*, pull-down assay; flotillin-1, *direct binding*, yeast two-hybrid system	[22]
	flotillin-1 and -2 (38 and 43 N-terminal a. a.), co-immunoprecipitation	
SNX4	flotillin-2 (SPFH domain), *direct binding*, in vitro pull-down assay	[46]
	flotillin-2, co-immunoprecipitation	
Rab11	flotillin-2 (SPFH domain), *direct binding*, in vitro pull-down assay	[46]
	flotillin-2, co-immunoprecipitation	
Hrs	flotillin-1, *direct binding*, in vitro pull-down assay	[12]
	flotillin-1 and -2, pull-down from cell lysates, co-immunoprecipitation	
Tsg101	flotillin-1 and -2, pull-down from cell lysates	[12]
BACE1	flotillin-1, *direct binding*, in vitro pull-down assay	[49]
	flotillin-1 and -2, pull-down from cell lysates; flotillin-2, co-immunoprecipitation	
EGF receptor	flotillin-1 and -2, co-immunoprecipitation	[9]
cRAF, MEK1, ERK2, cRAF, MEK1/2, ERK1/2, KSR1	flotillin-1, *direct binding*, in vitro pull-down assay	[9]
	flotillin-1, pull-down from cell lysates	[9]
γ-catenin	flotillin-1 and -2, *direct binding*, in vitro binding assay	[67]
	flotillin-1and -2, pull-down from cell lysates, co-immunoprecipitation	
polycystin-1, β-catenin, E-cadherin	flotillin-2, co-immunoprecipitation	[61]
N-cadherin, E-cadherin	flotillin-1 and -2, co-immunoprecipitation	[68]
Exo70, Fyn, ERK1/2, N-cadherin	flotillin-2, co-immunoprecipitation	[19]
CAP, ArgBP2, ArgBP2	flotillin-1, pull-down from cell lysates	[32]
CAP, Cbl	flotillin-1 (and not specified), pull-down from cell lysates, co-immunoprecipitation	[69,70,71]
PrP, Thy-1, Fyn, Lck	flotillin-1 and -2, co-immunoprecipitation	[18]
NPC1L1	flotillin- 1 and -2, co-immunoprecipitation	[72]
α-actinin	flotillin-1 and -2, pull-down from cell lysates, co-immunoprecipitation	[25]
Vav	flotillin-2, co-immunoprecipitation	[14]
LRAP	flotillin-1, pull-down from cell lysates, co-immunoprecipitation	[44]
syndecan-1	flotillin-1 (10-36 a.a), co-immunoprecipitation	[41]

BACE1, β-secretase 1; CAP, CBL-associated protein; Exo70, exocyst complex component Exo70; KSR1, kinase suppressor of Ras1; LRAP, leucine-rich amelogenin peptide; MEK1/2, mitogen activated protein kinase kinase 1/2; NPC1L1, Niemann-Pick C1-like protein 1; PrP, prion protein.

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
