# Peer review of "Flotillins: At the Intersection of Protein S-Palmitoylation and Lipid-Mediated Signaling"

_ijms, 2020, doi:10.3390/ijms21072283_

Round 1
Reviewer 1 Report
The authors wrote a review concerning the flotillins and their association with membranes, and the cellular processes linked to these associations. The review is well-documented and illustrated by comprehensive figures and detailed tables which synthesize existing literature data. The role of acylation and oligomerization of flotillins in relation to membrane interaction is well-described as well as the role of flotillins in signaling including signaling involving lipids. A summary concerning the presence/role of flotillins within extra-cellular vesicles is missing and could be added.
Figures
Figure 1 Specify in the legend the different color for myristoylation and palmitoylation.
Figure 2 Specify in the legend what are the green large bar and sphere.
Author Response
Reviewer I:
The authors wrote a review concerning the flotillins and their association with membranes, and the cellular processes linked to these associations. The review is well-documented and illustrated by comprehensive figures and detailed tables which synthesize existing literature data. The role of acylation and oligomerization of flotillins in relation to membrane interaction is well-described as well as the role of flotillins in signaling including signaling involving lipids.
A summary concerning the presence/role of flotillins within extracellular vesicles is missing and could be added.
- We have added this interesting subject in the revised manuscript (lanes 544-566) providing a description of the flotillin presence in extracellular vesicles and also of their potential involvement in exosome formation. We have also modified Table 1 to include the added data.
Figures:
Figure 1: Specify in the legend the different color for myristoylation and palmitoylation.
- We have marked the two types of acylation with green and red zigzags, as indicated in the revised Figure 1 legend. We have also added this explanation to the legends of Figures 2 and 3.
Figure 2. Specify in the legend what are the green large bar and sphere.
- Green rectangles and spheres indicate, respectively, transmembrane and peripheral membrane proteins, as indicated in the revised Figure 2 legend.
Reviewer 2 Report
The manuscript by Kwiatkowska et al. entitled “Flotillins: at the intersection of protein S-palmitoylation and lipid-mediated signaling” is a nice example of a clear and up-to-date review focused on a variety of physiological aspects of flotillins. Most recently, flotillins has captured the interest and attention of many researchers, as these ubiquitously expressed proteins appeared to be involved in a broad array of cellular processes, including cell signaling, endocytosis, trafficking etc. The authors made huge effort to discuss all the most important issues in the field, trying to build a concise picture. The manuscript is well written and accessible to a broad audience. Therefore I recommend its publication. Some minor concerns are as follows:
- OptiPrep is a trademark and refers to a solution of iodixanol
- correct form is “hematopoietic” not “hematopoetic”
- the abbreviation “zDHHC” is not properly defined
- “fluid phase” in Table 1 should be defined more unambiguously
- in general, the quality of English is fine, although minor check is required (e.g. inappropriate use of “an” in line 172)
- there are unnecessary gaps in the text (i.e. lines: 237, 264, 425, 475, 571, 715)
Author Response
The manuscript by Kwiatkowska et al. entitled “Flotillins: at the intersection of protein S-palmitoylation and lipid-mediated signaling” is a nice example of a clear and up-to-date review focused on a variety of physiological aspects of flotillins. Most recently, flotillins has captured the interest and attention of many researchers, as these ubiquitously expressed proteins appeared to be involved in a broad array of cellular processes, including cell signaling, endocytosis, trafficking etc. The authors made huge effort to discuss all the most important issues in the field, trying to build a concise picture. The manuscript is well written and accessible to a broad audience. Therefore I recommend its publication. Some minor concerns are as follows:
OptiPrep is a trademark and refers to a solution of iodixanol
- We have added this information in lane 248.
Correct form is “hematopoietic” not “hematopoetic”
- Corrected in lanes 320, 324.
The abbreviation “zDHHC” is not properly defined
- We have corrected the definition to “zinc finger and Asp-His-His-Cys domain-containing” in lane 91 and in the list of abbreviations.
“Fluid phase” in Table 1 should be defined more unambiguously
- We have now specified in Table 1 that it was the uptake of dextran, either magnetic or AF488-labeled.
In general, the quality of English is fine, although minor check is required (e.g. inappropriate use of “an” in line 172)
- The manuscript has been checked by a native English speaker. Several typos have been corrected and minor amendments to improve style, including adding or removing commas, introduced. As to the phrase in question (an S-palmitoylation”) we believe it is legitimate (pronounced es-palmitoylation hence “an” rather than “a”) and prefer not to change it. The same concerns “an SPFH domain” in lane 46.
There are unnecessary gaps in the text (i.e. lines: 237, 264, 425, 475, 571, 715)
- Excessive spaces have been removed.